# Molecular Mechanisms Linking Inflammation to Autoimmunity in Sjögren’s Syndrome: Identification of New Targets

**DOI:** 10.3390/ijms232113229

**Published:** 2022-10-30

**Authors:** Margherita Sisto, Domenico Ribatti, Sabrina Lisi

**Affiliations:** Department of Translational Biomedicine and Neuroscience (DiBraiN), Section of Human Anatomy and Histology, University of Bari “Aldo Moro”, Piazza Giulio Cesare 1, I-70124 Bari, Italy

**Keywords:** Sjögren’s syndrome, autoimmunity, apoptosis, aquaporin, angiogenesis, EMT

## Abstract

Sjögren’s syndrome (SS) is a systemic autoimmune rheumatic disorder characterized by the lymphocytic infiltration of exocrine glands and the production of autoantibodies to self-antigens. The involvement of the exocrine glands drives the pathognomonic manifestations of dry eyes (keratoconjunctivitis sicca) and dry mouth (xerostomia) that define sicca syndrome. To date, the molecular mechanisms mediating pathological salivary gland dysfunction in SS remain to be elucidated, despite extensive studies investigating the underlying cause of this disease, hampering the development of novel therapeutic strategies. Many researchers have identified a multifactorial pathogenesis of SS, including environmental, genetic, neuroendocrine, and immune factors. In this review, we explore the latest developments in understanding the molecular mechanisms involved in the pathogenesis of SS, which have attracted increasing interest in recent years.

## 1. Introduction

Sjögren’s syndrome (SS) is a multifactorial systemic autoimmune disease, the pathophysiology of which has not yet been fully deciphered [1,2] (Figure 1). SS is characterized by a wide spectrum of clinical manifestations and marked exocrine gland dysfunction. SS is classified as primary SS (pSS) when the clinical manifestations occur alone or as secondary SS when associated with another autoimmune disease [1,2].

Classically, it has been postulated that sicca symptoms in SS patients are a two-step process whereby the lymphocytic infiltration of the lacrimal and salivary glands (SG) is followed by epithelial cell destruction, resulting in keratoconjunctivitis sicca and xerostomia [3]. Recently, great efforts have been made to elucidate the mechanisms involved in the pathogenesis of the disease in order to identify potential new therapeutic targets in SS (Table 1). In recent years, interesting discoveries have shown that pSS has pathogenic mechanisms and etiology in common with other autoimmune diseases that predominantly afflict women, represented by rheumatoid arthritis (RA) and systemic lupus erythematosus, which preferentially affect specific target organs. Indeed, these autoimmune diseases, characterized by a chronic inflammatory condition, show similar clinical manifestations, serological profiles, and immunological alterations. Currently, the term poly-autoimmunity is used to indicate the co-existence of these three pathologies in the same patient, and, sometimes, these conditions can also be manifested by members belonging to the same family. This suggests that the molecular mechanisms underlying the onset of these pathologies could be the same, and elucidating these mechanisms in pSS could be of help in guiding researchers toward understanding the pathogenesis of related autoimmune diseases [4].

In this review, we provide a summary of the recent literature on the discoveries of new molecular mechanisms in SS SGs, focusing on apoptosis, angiogenesis, aquaporins, and epithelial–mesenchymal transition (EMT). These processes represent recent and very promising research fields from a therapeutic point of view that could help to individuate new avenues for novel treatment options.

## 2. Recent Advances in Apoptosis in SS

Over the last several years, an increasing number of studies have revealed the key role of apoptosis, or programmed cell death (PCD), in the pathogenesis of SS [5,6]. Apoptosis is a critical process, highly complex and sophisticated, that is conserved throughout evolution, development, and aging to ensure both physiological and morphological changes as well as the elimination of damaged cells [7]. The mechanism of apoptosis is implicated in a cascade of molecular events that lead to the development of autoimmune disorders [8]. In the last few decades, apoptosis has been hypothesized as a mechanism of cell death in the SGs of pSS patients on the basis of data collected using experimental SS mouse models [9], pointing to glandular epithelial cells as active players in this mechanism [10]. In fact, pSS patients present increased apoptosis in the salivary glandular epithelium and show the co-localization of Fas (Apo-1/CD95) antigen and Fas ligand (FasL) in ductal and acinar cells, suggesting that the trigger of the apoptotic cascade occurs through the Fas/FasL system [11]. On the other hand, cytotoxic T cells have also been reported to play a crucial role in apoptotic events in the SS salivary epithelium. Indeed, cytotoxic T cells surround the epithelial cells following inflammatory responses and, through Fas ligand interactions and perforin and granzyme release, promote apoptosis in the pSS epithelium, thus leading to advanced glandular tissue destruction and chronic sialadenitis [12,13].

It is reported that there are multiple pathways involved in the apoptotic mechanism in SS. One of the best-known pathways that determine cell fate and evolution toward apoptosis is mediated by caspase cascade activation via the extrinsic or intrinsic mechanism (Figure 2). Caspases cleave a variety of protein substrates within the cell, including alpha-fodrin [14,15] and Ro and La proteins [5,16], and, therefore, it is conceivable that epithelial cell apoptosis may provide cellular proteins as autoantigens that then perpetuate the autoimmune response in SS [17,18,19].

### 2.1. Pyroptosis

In the field of SS apoptosis research, recently, the excessive exacerbation of pyroptosis has been proven to play a crucial role. Pyroptosis is an overactive PCD characterized by pore formation in cell membranes, cell rupture, and the release of intracellular contents and pro-inflammatory cytokines, such as IL-1β and IL-18. This prolonged release of inflammatory factors drives the overactivation of the immune system, promoting autoimmunity [20]. The discovery of the activation of pyroptosis in SS creates a connecting bridge with the activation of cytotoxic T cells observed in SS, as, under this condition of the uncontrolled release of pro-inflammatory cytokines, autoantibodies and/or autoreactive T cells can uncontrollably attack the body, causing autoimmune diseases [21]. Among the canonical inflammasomes that enable the induction of pyroptosis (NLRP1, NLRP3, NLRC4, interferon-inducible protein AIM2, and pyrin) [22], the increased expression of NLRP3 (NOD (nucleotide oligomerization domain)-, LRR (leucine-rich repeat)-, and PYD (pyrin domain)-containing protein 3) inflammasome-related elements in peripheral blood mononuclear cells or macrophages infiltrating the SGs of pSS patients was detected, determined by inflammatory circulating cell-free DNA accumulated in the SS patients’ serum [23]. Similarly, the accumulation of damaged cytoplasmic DNA in the SG ductal cells of patients with pSS was demonstrated to activate the AIM2 inflammasome, causing the intensive expression of pyroptosomes in the SG tissue [24]. Pyroptosis is closely associated with the activation of caspase-dependent cascades, and, in SS, type I IFN upregulated the expression of caspase-1 in pSS epithelial cells (SGECs) and may accelerate NLRP3 or AMI2 inflammasome-associated pyroptosis [25].

### 2.2. Apoptosis and Viral Infection in SS 

Interestingly, pyroptosis, driving CD4 T-cell depletion in HIV-1 infection [26], could be one of the molecular mechanisms involving viral proteins in the etiopathogenesis of pSS [27]. The role of viral infections in the pathogenesis of SS represents an important line of research carried out by the Nakamura group, demonstrating that viruses could essentially change the expression or regulation of various genes, including those that regulate apoptosis [28]. In pSS, the initial apoptosis of epithelial cells may be a normal response to viral infection, but the inability to regulate the apoptotic process may then perpetuate epithelial cell impairment and the resultant cellular and humoral features that characterize SS. In fact, viral infections, such as Epstein–Barr virus (EBV) and human T-cell leukemia virus type 1 (HTLV-1), change the phenotype or features of SS SGECs through the breakdown of local immunological tolerance and determine the activation of the apoptotic cascade [28].

Indeed, there are recent insights regarding the relationship between apoptosis and viral infection in SS. Increased retinoblastoma-associated protein 48 (RBAp48) expression, which controls chromatin organization induced by viral infections such as HIV (human immunodeficiency virus), was also found in the epithelial cells of minor SGs of labial biopsies from patients affected by pSS, in which the cells undergo the apoptotic process [29]. Therefore, the RBAp48 of SGs cells in mice and humans was reported to be upregulated by estrogen deficiency, which triggered apoptosis in the target cells [29,30,31]. Although the relationship between viral infection and autoantigen formation in SS is shrouded in mystery, apoptosis in target cells due to viral infection can induce the upregulation of multiple enzymatic activities to generate pathogenic epitopes from intracellular molecules, leading to an autoimmune response [31].

### 2.3. Lysosome-Associated Membrane Protein 3-Dependent Apoptosis

Another crucial point in the pathogenesis of SS is the upregulation of lysosome-associated membrane protein 3 (LAMP3), a membrane glycoprotein predominantly localized in lysosomes induced by IFN. An interesting recent study [32] demonstrated the increased expression of LAMP3 in a subset of pSS cases. The stratification of patients based on their clinical features suggested a link between increased LAMP3 expression and the presence of serum autoantibodies, including anti-Ro/SSA, anti-La/SSB, and anti-nuclear antibodies. [32,33]. In vitro findings showed that the transfection of LAMP3 expression plasmids in cultured SGECs triggered caspase-3 activity that leads to the apoptotic process [32,33]. Therefore, additional studies in vivo have demonstrated that the increased expression of LAMP3 provokes apoptosis in SGECs derived from non-obese diabetic (NOD) mice, a well-characterized model of the spontaneous onset of an SS-like phenotype, in which LAMP3 is locally overexpressed in the submandibular glands [33].

A schematic representation of the mechanisms described in the above paragraph is shown in Figure 2.

However, although many of the key apoptotic proteins that are activated or inactivated in apoptotic cascades have been discovered, the molecular events of the action or activation of these proteins in SS are not fully understood and are the focus of continued research. 

## 3. Angiogenesis in SS

Angiogenesis is a fundamental process in growth, development, and repair [34]. Besides its well-known role in cancer, it has become clear that angiogenesis is also a critical component of non-neoplastic chronic inflammatory and autoimmune diseases, including atherosclerosis, RA, diabetic retinopathy, psoriasis, airway inflammation, peptic ulcers, Alzheimer’s disease, and SS [35,36]. In chronic inflammation, angiogenesis mediates the expansion of the microvascular tissue bed through the activation and proliferation of endothelial cells, leading to capillary and venule remodeling [37]. The expansion of the microvascular bed determines, in turn, the recruitment of inflammatory cells; for this reason, angiogenesis and inflammation seem to be chronically co-dependent processes [37]. 

### 3.1. Neo-Angiogenesis in pSS SGs

Neo-angiogenesis is mediated by the activity of vascular endothelial growth factor-A (VEGF-A) and its main receptor, vascular endothelial growth factor receptor-2 (VEGFR-2) [35,36,37,38]. In recent years it has emerged that, in pSS, infiltrating T cells and human SGECs produce increased amounts of pro-angiogenic factors via VEGF-A/VEGFR-2 system activation; consequently, VEGFR-2 blockade could be an entirely novel approach to blocking experimental angiogenesis and inflammation in pSS [39,40,41]. This agrees with the observation that the physiological response to tissue injury or infection is an increase in vascular permeability, and the microvascular changes associated with angiogenesis are key contributors to the tissue damage and remodeling processes that inevitably accompany chronic inflammation [39,40,41]. However, to date, few data are available on the role of angiogenesis in SS. Published data demonstrate that a functional impairment of the arterial wall may sustain early phases of atherosclerotic damage in pSS, and correlated chronic inflammation and immunological factors appear to be involved in the dysfunction of endothelial and vascular smooth muscle cells [40,41]. Confirming these hypotheses, strong positive staining for VEGF-A and VEGFR-2 proteins in pSS SG biopsies was detected [38], and a great number of pro-angiogenic proteins are overproduced in pSS SGs [38,40]. Recent research has been enriched by the placement of neuropilin in this scenario, a transmembrane co-receptor for members of the VEGF family, first described as mediators of neuronal organization, which seems to promote angiogenesis in pSS through the activation of NF-κB. [42]. However, direct evidence of neo-angiogenesis in SS is still lacking, although the sprouting of new vessels from preexisting ones was detected, favoring active macrophage and histiocyte infiltration [41]. The vasculature of SGs plays a critical role in saliva secretion [41], and the analysis of the changes that occur in the blood vessels during the progression of SS could help the understanding of the hyposalivation symptom. By virtue of its great therapeutic potential, in recent years, further investigations have been carried out in this field, leading to discoveries that are not always concordant. McCall and colleagues [43] showed decreased VEGF expression levels in pSS SGs, along with similar blood and lymphatic vessel organization and volume fractions, suggesting that angiogenesis and lymphangiogenesis do not play a significant role in the progression of SS. These conclusions were made with the use of selective markers for endothelial cells.

### 3.2. Endothelial T Cells in pSS

On the contrary, experimental data collected by Alunno and colleagues demonstrated that endothelial dysfunctions were present in pSS patients, represented by abnormalities in the endothelial cell structure that allow favorable conditions for the formation of atherosclerotic plaques [44]. In physiological conditions, endothelial cells may be damaged by several stimuli, including shear stress and transmural pressure, but they are promptly replaced thanks to the release of endothelial progenitor cells (EPC) from the bone marrow, which migrate to the site of injury and undergo a full maturation process. The assessment of circulating EPC along with circulating endothelial microparticles (EMP), which act as surrogate biomarkers of endothelial dysfunction, allowed researchers to verify that this process occurs in SS [45]. In particular, an increase in EPC in parallel to an increase in EMP may suggest a compensatory mechanism to overcome endothelial cell damage [45]. 

Therefore, in recent years, another leading actor in the scenario of endothelial repair has been identified, so-called angiogenic T cells (Tang) characterized by the co-expression of CD3, CD31, and CXCR4 [46]. Although consistent endothelial damage is ongoing in pSS, as proven by an increased amount of circulating EMP compared to healthy controls, a counteracting mechanism leading to EPC release from the bone marrow also takes place [45]. The current observations that circulating Tang cells are also raised in pSS, that they are significantly correlated to their partner EPC, and that both Tang and EPC are significantly associated with the EULAR SS disease activity index (ESSDAI) unmask another facet of this complex process. However, this makes it even more difficult to understand why, although the endothelial repair machinery seems to be fully working, pSS patients still display higher cardiovascular risk, and those with higher disease activity are at even more risk than those with milder disease [46]. Furthermore, Tang cells, which are numerous and close to blood vessels in pSS MSG, can produce IL-17, a cytokine that was recently demonstrated to play a versatile role in the pathogenesis of SS [47]. A hypothetical scenario showing the recent advances in SS angiogenesis is reported in Figure 3.

### 3.3. Possible Occurrence of Atherosclerosis in SS

Other authors have deepened the understanding of the contribution of chronic inflammation and neo-angiogenesis to precocious atherosclerosis in patients with SS. Microvascular disease is a hallmark of SS, in which the dysregulation of vascular tone, altered and defective angiogenesis, and endothelial cell injury or activation due to a chronic inflammatory condition have been detected [48]. Despite this evidence, the direct role of the accumulation of pro-inflammatory cytokines in driving atherosclerosis in SS has not been fully elucidated. On the other hand, the increased expression of endothelial damage markers in pSS could suggest that other factors or concomitant mechanisms could be involved in the increased cardiovascular (CV) risk in SS. It is interesting to note that patients with systemic extra-glandular involvement, including the central nervous system, show a higher risk of CV events, and, on the other hand, patients affected by CV risk factors present a higher prevalence of extra-glandular manifestations, often involving central nervous system [45]. Enhanced levels of circulating EMP were detected in a pSS cohort, especially in patients with a long pSS disease duration, suggesting a chronic endothelial injury in which it is no longer possible to carry out a repair due to the impoverishment of the reserve of endothelial progenitors [45]. Endothelial activation in SS has also been suggested by the detection of other specific markers that are increased in patients in comparison with healthy subjects, such as soluble thrombomodulin (TM) [49] and nitrotyrosine [40]. TM, a membrane glycoprotein that binds to the high-affinity receptor for thrombin on the endothelial cell membrane, acting on fibrinolysis and coagulation, promotes atherosclerosis through its mitogenic activity in vascular smooth muscle cells; soluble TM reflects the degree of endothelial cell damage, which has a direct role in atherosclerotic lesions. In addition, the detection of the increased expression of nitrotyrosine-modified proteins in both plasma and atherosclerotic lesions from pSS patients with concomitant coronary artery disease and in an experimental mouse model of atherosclerosis suggests that oxidative stress may be a concomitant trigger of atherosclerosis in SS; this consideration derives from the fact that protein nitration is a post-translational modification caused by reactive nitrogen species, and the interaction between oxidative stress and inflammatory factors may act synergistically in the onset of atherosclerosis in SS [50]. Among the other factors detected at high levels in SS that could be implicated in the possible activation or alteration of endothelial cells are the soluble adhesion molecules ICAM-1 and VCAM-1 described in a pSS cohort, which could be indicative of chronic endothelial damage consequent to the subintimal infiltration of inflammatory cells within the arterial wall, once again underlining the close association between the neo-angiogenesis or alteration of the endothelium and chronic inflammation [40] (Figure 3). The understanding of the role of angiogenesis in the pathogenesis of SS requires further studies to provide data that will facilitate the development of novel molecular therapeutic targets. 

## 4. Novel Insights into the Role of Aquaporins in the Pathogenesis of SS

The mechanisms responsible for SG hypofunction and the corollary of hyposalivation are not fully deciphered, but recent publications have underlined the involvement of aquaporins (AQPs) in the molecular and cellular mechanisms underlying SS pathogenesis [51]. AQPs seem to be potential unexpected actors in sicca syndrome. AQPs are part of a family of 13 transmembrane channel proteins implicated in transcellular water permeability that have been discovered in all living organisms [52]. Of the known AQPs, six are expressed in mammalian SGs [53,54]. Emerging data reveal a new function of AQPs in the inflammatory process, as demonstrated by their dysregulation in a wide range of chronic inflammatory diseases, including SS [55,56]. Indeed, the expression, localization, and function of AQPs have been extensively studied in the SGs of patients affected by SS [55,56]. In SG biopsies from SS patients, AQP1 expression was considerably decreased in myoepithelial cells (MECs) as compared to healthy controls [57]. This diminished AQP1 expression in MECs raised the hypothesis that the alteration of the myoepithelial cell may play a key role in decreased salivary secretion in SS disorder [57]. Recently, the expression of the AQP4 gene and the distribution of the AQP4 protein was demonstrated in healthy control glands and pSS SG biopsies to be confined, in particular, to the basal region of acini and to the lateral and apical membranes of intercalated and striated ducts [58]. Interestingly, Sisto et al. also observed the membrane localization of the AQP4 protein in MECs [59] in both healthy and pSS SGs. The novelty of this study was the detection of AQP4 at lower levels in pSS MECs in comparison with healthy controls. Thus, the marked downregulation of AQP4 expression in pSS MECs suggests that these cells do not function normally in pSS SGs; some activities of salivary secretion may be impaired, and thus, the altered functionality of these cells could be responsible for the loss of saliva secretion in these patients [59]. Additionally, the defective localization of AQP5 has been extensively demonstrated in pSS patients, with its localization mainly distributed at the basolateral membrane rather than at the apical membrane of acinar cells. Dysregulated AQP5 expression has been proposed to contribute to the loss of saliva secretion observed in SS patients [60]. 

Interestingly, several compelling lines of evidence underline a link between inflammation and altered AQP expression; in particular, dysregulated AQP5 expression is often described in association with a condition of chronic inflammation [61,62]. Among the long list of cytokines that have been implicated in several molecular pathogenic mechanisms in SS, some have been closely associated with altered AQP5 expression/localization. Indeed, studies were performed to assess whether IL-7 could, in some way, be connected with altered AQP5 localization in SG [63]. Promising results were obtained, because, by blocking IL-7 production, a reduction in SG inflammation was observed, along with an improvement in functionality [63]; using an experimental mouse model of SS, an IL-7 block led to restored levels of AQP5, improving the manifestations of SS [63]. In addition, the injection of anti-TNF antibodies into NOD mice reduced SG inflammatory foci and increased AQP5 protein expression [64]. 

More recently, various antibodies against several AQPs have been detected in the sera of patients suffering from SS. Autoantibodies against AQP5 were detected in a cohort of pSS patients and were related to significantly lower basal salivary secretion levels [65]. The role of these autoantibodies against AQP5 has been recently deciphered, demonstrating that these autoantibodies bind to the extracellular loops of AQP5 and obstruct water flux transit [65]. 

This observation could explain the role of AQP5 autoantibodies in the pathogenesis of SS or sicca syndrome. Furthermore, other autoantibodies against AQP1, AQP3, AQP8, and AQP9 have been revealed in patients with pSS [66], which could be responsible for the alteration of the flow of water in SGs as the basis of the reduced capacity for secretion. Advances in AQP research in SS are schematically reported in Figure 4. 

Further studies are necessary to identify promising new approaches for the restoration of SG function through AQP involvement in SS. In fact, the modulation of AQP gene and protein expression in SS patients could be a potential therapeutic option for re-establishing SG activity and alleviating the disadvantage caused by sicca syndrome.

## 5. New Insights into the Regulation of Epithelial–Mesenchymal Transition in SS

Fibrosis is often present as the final common pathological outcome of most chronic inflammatory disorders, and, when organs are injured, a sequence of pleiotropic signaling activates the immune system, leading to the exacerbation of inflammation and thereby triggering EMT. EMT is a complex reprogramming process in which epithelial cells gradually shift toward mesenchymal-like cells [67,68,69,70,71]. Recently, it has been demonstrated in SGs that chronic injury often triggers the EMT cascade, leading to severe fibrosis and the atrophy of organs [72,73,74,75]. Therefore, the fibrotic process compromises the secretory activity of SGs, leading to hyposalivation [72]. Recently, Leehan’s group developed a digital measurement of SG fibrosis, demonstrating a pathological contribution of interstitial fibrosis to the development of SG hypofunction, clarifying the molecular changes involved in severe fibrosis evolution in SS [72].

It is now widely unquestioned that the development of a fibrotic program in SS is due to the production of fibrogenic mediators by inflammatory and epithelial cells; among these mediators, a main role is played by the growth factor TGF-β1 [76]. Pioneering studies conducted by Sisto and colleagues demonstrated that TGF-β1 promotes an SG epithelial cell transition toward the mesenchymal phenotype through the activation of EMT-dependent fibrosis [75,77,78]. Experimental in vitro studies performed on human SG epithelial cells have shown that TGF-β1 induces the transition of SG epithelial cells from the classic cobblestone morphology toward a more spindle-shaped fibroblast cell type characterized by a reduction in the expression level of epithelial adherence junction proteins [75,77,78]. This was supported by the observation that SG biopsies derived from pSS patients show an elevated expression of TGF-β1 [77,78]. In addition, Koski et al. reported compelling evidence for the marked expression of TGF-β family member proteins, including TGF-β2, in the fibrotic area of pSS SGs [76]. The altered upregulation of TGF-β1 in the pSS SGs induces the EMT process via the triggering of small mothers against decapentaplegic (SMAD)-mediated canonical and SMAD-independent non-canonical pathways [75]. Confirming these observations, recent studies have demonstrated the involvement of the TGFβ1/SMAD/Snail (Zn finger protein) signaling pathway in pSS SG fibrotic transformation, as confirmed by the revelation of the notable distribution of TGF-β1, pSMAD2/3, and SMAD4 proteins in pSS SG tissues [75]. Furthermore, more interestingly, pSS SGs were demonstrated to show strong positivity for EMT-characterizing factors, such as α-SMA (α-smooth muscle actin), vimentin, and collagen type I, and, at the same time, the downregulation of the cell–cell adhesion protein E-cadherin, considered an epithelial marker [75,77,78].

To date, attempts to explain the development of the fibrotic scenario in SS SGs induced by the EMT process have focused on the crucial role of various pro-inflammatory cytokines. The results are very promising; Sisto et al. discovered that IL-17 and IL-22 participate in the triggering of TGF-β1/EMT-dependent SG fibrosis, depending on the inflammatory grade of the disease [75].

Interesting results were also obtained by testing the pro-fibrotic activity of IL-6, detected at very high levels in pSS SGs, which allowed researchers to prove that IL-6 determines the activation of EMT-dependent fibrosis in pSS SGECs, accompanied by increased levels of vimentin and collagen type I and reduced E-cadherin gene expression [77,78].

Recent studies have observed that epithelial environmental changes induce EMT-associated gene transcription, such as the BMP7 (Bone Morphogenetic Protein 7) gene, as well as the WNT gene family, initiating the cellular signaling circuitry that regulates EMT [79]. In support of these findings, the transcription of these genes is thought to negatively modulate epithelial marker expression while driving the increased expression of markers associated with the fibroblastic state [79].

Actually, interestingly, BMP signaling is considered an important inducer of EMT, which can be initiated by chronic inflammation. Therefore, BMP proteins often serve as antagonists of TGF-β, and a study demonstrated that both BMP3 and TGFB2 expression levels are increased in pSS patients with severe SG fibrosis; in accordance with these findings, BMP6 is also involved in SG hypofunction [79].

Alterations in the distribution and expression of metalloproteinases (MMPs) have also often been considered in the induction of the EMT process. Interestingly, a recent paper established the role of ETS1 (ETS Proto-Oncogene 1, Transcription Factor) as a driver of pathological MMP9 overexpression in the salivary epithelium of SS patients. This mechanism triggers the expression of EMT factors responsible for the severity degree of SG damage and for the morphological and functional changes contributing to SG fibrosis [80].

The study of the molecular mechanisms underlying the activation of the EMT process has been enriched in recent years by new players, which will then need to be placed in the right position within the complex scenario of this cellular transformation process. Follistatin is one of the most recent candidates to fill this role; follistatin-like 1 protein (FSTL1), a secreted glycoprotein that has been shown to participate in regulating developmental processes, was demonstrated to be produced primarily by cells with the mesenchymal phenotype in the EMT process [81]. Interesting recent studies have underlined that FSTL1 promotes EMT in concert with TGF-β1 [82,83]. Original data emerging in the last few months have shown that FSTL1 is upregulated in pSS and is able to activate the SMAD2/3-regulated TGF-β1-dependent EMT cascade in SS, promoting fibrogenesis [84]. A schematic representation of the complex EMT activation in SS is shown in Figure 5.

## 6. Conclusions

Multiple immunomodulators are involved in the pathogenesis of SS, including autoantigens, inflammatory cells, cytokines, and chemokines, which have a contributory role in the multiple and complex molecular mechanisms responsible for the SG atrophy and hyposalivation characterizing SS. Despite the complex disease etiology and our incomplete understanding of the relationship between these factors and SG dysfunction, we presume that multifactorial situations affect immune dysregulation and influence SS development. Having considered pioneering research fields, the aim of this review was to provide food for thought that can lead scientists to investigate possible commissions between the various latest intricate mechanisms analyzed in the hope of identifying, in the future, a single general mechanism that determines the onset of SS syndrome. Understanding the underlying mechanisms may shed light on SS pathogenesis, as well as new and effective strategies for the management of this disease.

## Figures and Tables

**Figure 1 ijms-23-13229-f001:**
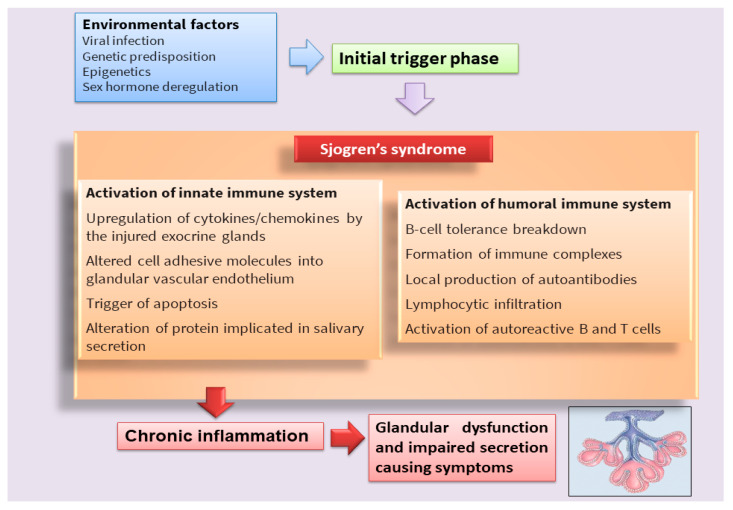
Scheme that clarifies the hypothetical onset of SS.

**Figure 2 ijms-23-13229-f002:**
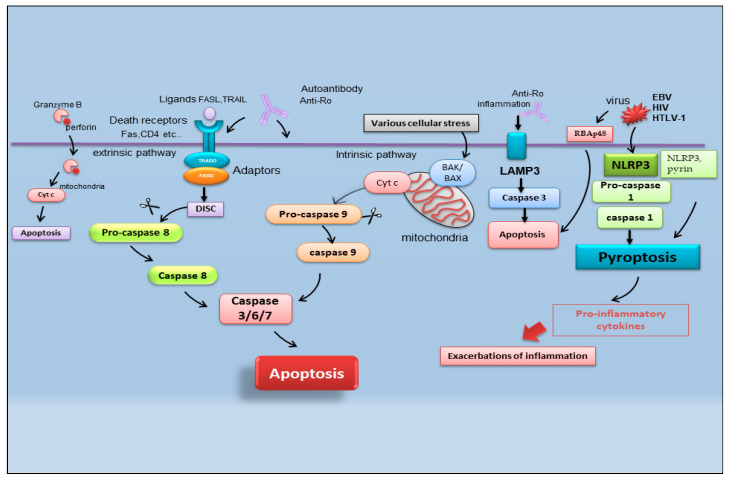
Schematic representation of recently identified apoptotic events in SS. B-cell lymphoma protein 2 antagonist killer 1 (BAK); B-cell lymphoma protein-2-associated X protein (BAX); cytochrome c (Cyt c); death-inducing signaling complex (DISC); Epstein–Barr virus (EBV); fatty acid synthetase ligand (FASL); Fas-associated protein with death domain (FADD); human immunodeficiency virus (HIV); human T-cell leukemia virus type 1 (HTLV-1); lysosome-associated membrane protein 3 (LAMP3); nucleotide-binding domain (NOD)-like receptor protein 3 (NLRP3); tumor necrosis factor receptor type 1-associated DEATH domain (TRADD); TNF-related apoptosis-inducing ligand (TRAIL) retinoblastoma-associated protein 48 (RBAp48).

**Figure 3 ijms-23-13229-f003:**
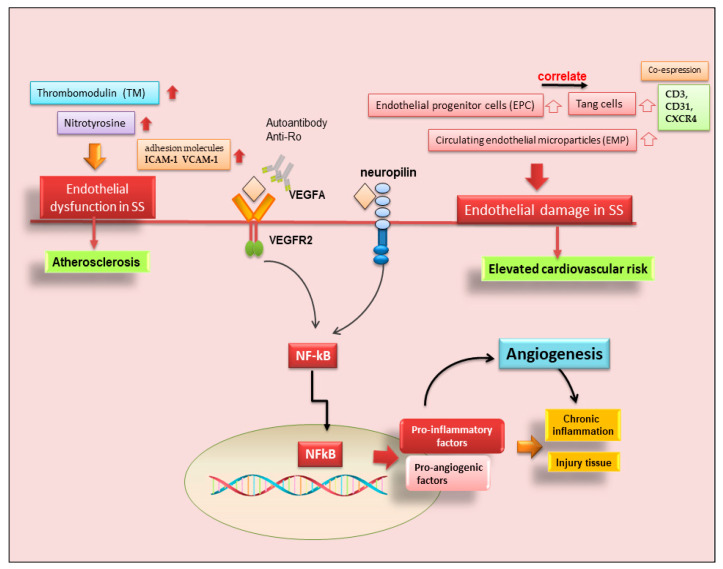
Role of angiogenesis in SS and SS-related diseases. Cluster of differentiation 3 (CD3); cluster of differentiation 31 (CD31); chemokine receptor type 4 (CXCR4); intercellular adhesion molecule 1 (ICAM-1); nuclear factor kappa B (NF-kB); vascular cell adhesion molecule 1 (VCAM-1); vascular endothelial growth factor A (VEGFA); vascular endothelial growth factor receptor 2 (VEGFR).

**Figure 4 ijms-23-13229-f004:**
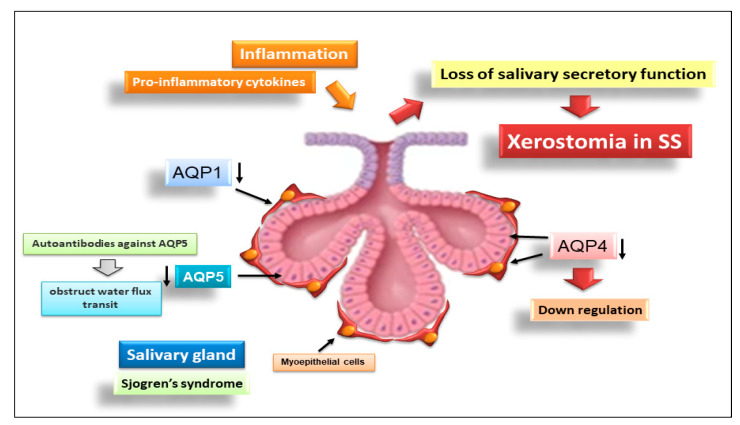
Advances in AQP research in SS (aquaporin, AQP).

**Figure 5 ijms-23-13229-f005:**
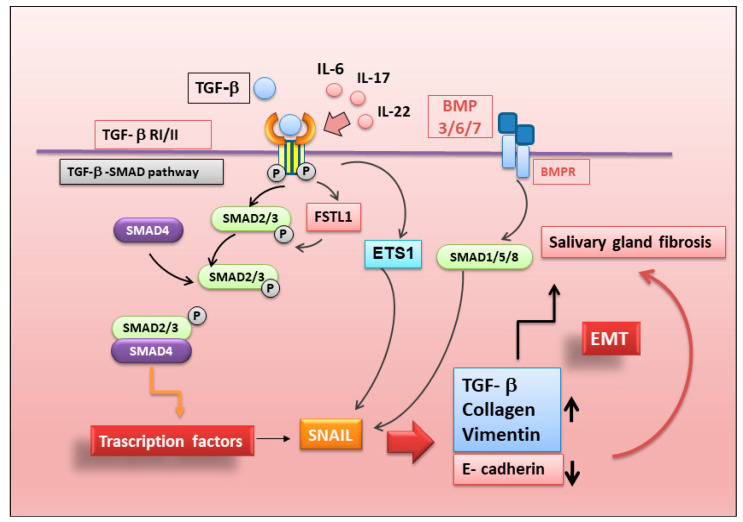
Possible mechanisms of EMT activation in SS. Bone Morphogenetic Protein (BMP); ETS Proto-Oncogene 1, Transcription Factor (ETS); Follistatin-Like Protein 1 (FSTL1); Interleukin-6 (IL-6); Interleukin-17 (IL-17); Interleukin-22 (IL-22); mothers against decapentaplegic homolog 2 (SMAD2), mothers against decapentaplegic homolog 3 (SMAD3); mothers against decapentaplegic homolog 4 (SMAD4); zinc finger protein SNAI (SNAIL); transforming growth factor beta (TGF-β); transforming growth factor-β type I receptor (TGF-β RI/I).

**Table 1 ijms-23-13229-t001:** Schematic illustration of therapies used in SS.

Nomenclatures of Therapies	Definition	Examples
Thopical therapies	Interventions directly applied to the mucosal surface involved	Saliva substitutes, ocular tears, ocular gels/ointment
Systemic therapies	Drugs administered orally or intravenously for systemic disease	Antimalarials, glucocorticoids, immunosuppressive agents, intravenous immunoglobulins, biologics drugs
Systemic therapies for severe refractory diseases	Drugs administered intravenously	B-cells targeted therapies

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
