# Peer review of "Molecular Mechanisms Linking Inflammation to Autoimmunity in Sjögren’s Syndrome: Identification of New Targets"

_ijms, 2022, doi:10.3390/ijms232113229_

Round 1
Reviewer 1 Report
General comments:
I think you should distinguish between SS and pSS (primary Sjögren´s Syndrome). If you mean pSS, please use that throughout the text (you have used pSS in some paragraphs). Or specify in the Introduction that you mean both pSS and secondary SS.
Please also distinguish between xerostomia (the sensation of oral dryness) and hyposalivation (the objective measure of reduced saliva secretion). A pSS patient can have hyposalivation without a sensation of xerostomia. And vice versa. If you mean hyposalivation, please use that word throughout.
And when it comes to the references, around 16 out of the 91 listed include yourselves. That is close to 20%, and seems to me a bit excessive. I would recommend that you amend your list and reduce your own contribution. You should also consider to include more than one reference for a finding (where possible).
Abstract:
Please revise this sentence: “While the scenario is clear and characterized by the 17 selective and progressive atrophy of the SGs, researchers view on the participation of the different actors has evolved over the years.” as it is not easy to understand what you mean by “the scenario is clear” and “different actors”.
Main document:
A great deal of details are presented in this manuscript. I would consider simplifying it a bit by reducing the number of molecules presented. Or at least try to present them in an orderly table as a supplement. The nice figures help some but they are not complete.
In general, the paragraphs are rather long. This is especially true for section 3. Please consider dividing them in some manner for easier readability.
Line 43: You cannot establish that “apoptosis has been identified as a mechanism of cell death in SGs of SS patients using experimental SS mouse models”. The information gained from mouse models cannot be used to “establish” a mechanism in humans. Please modify.
Line 48: Please correct “cytotossic T cells”.
Line 86: Please rewrite as this statement is not clear: “thus as the genes that regulated apoptosis process”.
Line 121: Please remove the unnecessary information.
Figures:
Some misspellings in figures. Please correct: Figure 1: mitochondria, exacerbations; Figure 2: Endothelial damage in SS. And I think the figure legends should include the abbreviations. Otherwise the reader will have to scan the text to find them all.
Author Response
We would like to express our sincere gratitude to the reviewer for her/his constructive and positive comments and for the very thoughtful critique of our manuscript and are pleased to say that we tried to address all the concerns raised. All changes to the manuscript are highlighted in the text. We respond below in detail to each of the reviewer’s comments and we hope that the reviewer will find satisfactory our responses to her/his comments.
General comments:
I think you should distinguish between SS and pSS (primary Sjögren´s Syndrome). If you mean pSS, please use that throughout the text (you have used pSS in some paragraphs). Or specify in the Introduction that you mean both pSS and secondary SS.
Thanks for your helpful comment. I have specified in the introduction the difference between primary and secondary SS and these terms have been appropriately reported in the manuscript.
Please also distinguish between xerostomia (the sensation of oral dryness) and hyposalivation (the objective measure of reduced saliva secretion). A pSS patient can have hyposalivation without a sensation of xerostomia. And vice versa. If you mean hyposalivation, please use that word throughout.
As you suggested, I have reported the term hyposalivation in the text, where necessary.
And when it comes to the references, around 16 out of the 91 listed include yourselves. That is close to 20%, and seems to me a bit excessive. I would recommend that you amend your list and reduce your own contribution. You should also consider to include more than one reference for a finding (where possible).
As suggested we have eliminated most of the references relating to studies conducted by our group.
Abstract:
Please revise this sentence: “While the scenario is clear and characterized by the 17 selective and progressive atrophy of the SGs, researchers view on the participation of the different actors has evolved over the years.” as it is not easy to understand what you mean by “the scenario is clear” and “different actors”.I agree with your comment and I changed the sentence trying to make it clearer.
Main document:
A great deal of details is presented in this manuscript. I would consider simplifying it a bit by reducing the number of molecules presented. Or at least try to present them in an orderly table as a supplement. The nice figures help some but they are not complete.
Following your suggestion, I tried to make the figures more complete and to specify the abbreviations used in the legends.
In general, the paragraphs are rather long. This is especially true for section 3. Please consider dividing them in some manner for easier readability.
I would like to specify that the topics covered in this review are complex and to summarize the data collected in recent years has sometimes proved difficult. However, I agree that paragraph 3, in particular, is very long, and it has been divided into 3 subparagraphs.
Line 43: You cannot establish that “apoptosis has been identified as a mechanism of cell death in SGs of SS patients using experimental SS mouse models”. The information gained from mouse models cannot be used to “establish” a mechanism in humans. Please modify.
Done
Line 48: Please correct “cytotossic T cells”.
Done
Line 86: Please rewrite as this statement is not clear: “thus as the genes that regulated apoptosis process”.
Done
Line 121: Please remove the unnecessary information.
Done
Figures:
Some misspellings in figures. Please correct: Figure 1: mitochondria, exacerbations; Figure 2: Endothelial damage in SS. And I think the figure legends should include the abbreviations. Otherwise the reader will have to scan the text to find them all.
Thank you for your comment which is very useful in improving understanding of the manuscript. All errors have been corrected and the legends have been rewritten by clarifying the abbreviations used.
Reviewer 2 Report
The review article by Margherita Sisto et al. explored the recent literature and the development focused on the discoveries of new molecular mechanisms involved in the pathogenesis of Sjögren’s syndrome (SS) that could help new avenues for novel treatment options and therapeutic approaches. The content in the review would be an addition to the currently available knowledge and literature. Overall, I think this review has the potential to be a very nice resource for those in the field and the authors do a nice job of walking readers through the recent updates. The manuscript is well-written. However, I did have a few comments about the current version, as detailed below.
· The review assumes a lot of knowledge on the part of the reader. It would be greatly improved if an expanded introduction and/or a new first section provides a more extensive overview of how the immune system goes awry to cause SS (ideally including a schematic) and what specific gaps exist in our understanding, diagnostics, and treatments.
· It's very common for individuals who have SS to also have a rheumatic disease such as rheumatoid arthritis or lupus, is there a common molecular or immunological association involved? The authors have not discussed/mentioned any single line about this.
· In general, it’s nice to end an introduction section with a few comments about what the review contains, to help readers understand what’s coming; please consider adding some relevant remarks here.
· It would be great to have a table with detailed information on the past and present therapies used, the mechanism targeted, outcomes, and limitations against SS.
· The schematics presentations by the authors made it attractive and easy for the readers to grasp.
· There are a lot of definite and indefinite articles missing and is required to correct these.
· Please correct the manuscript for language and grammatical mistakes.
Author Response
We thank the reviewer for having judged our manuscript very positively and for the very thoughtful critique of our manuscript and are pleased to say that we tried to address all the concerns raised. All changes to the manuscript are highlighted in the text. We respond below in detail to each of the reviewer’s comments and we hope that the reviewer will find satisfactory our responses to his comments.
The review assumes a lot of knowledge on the part of the reader. It would be greatly improved if an expanded introduction and/or a new first section provides a more extensive overview of how the immune system goes awry to cause SS (ideally including a schematic) and what specific gaps exist in our understanding, diagnostics, and treatments.
Thanks for your suggestion, effectively a short introduction is needed to clarify the main characteristics of Sjogren's syndrome and its onset. For this reason, we have added in the introduction a scheme (new figure 1) that clarifies how a dysregulation of the immune system can determine the onset of SS syndrome and a generic table related to SS therapeutic strategy; these new figures can help the reader in understanding the disease object of this review. I would like to clarify that it was not possible in this context to reuse in a more specific way the past and future therapies used for SS patients because it is a topic too extensive to be summarized in this context.
It's very common for individuals who have SS to also have a rheumatic disease such as rheumatoid arthritis or lupus, is there a common molecular or immunological association involved? The authors have not discussed/mentioned any single line about this.
Thank you for this interesting comment; I mentioned in the introduction the possibility that SS disease is linked to other autoimmune diseases having in common molecular mechanisms and aetiology; a related very recent reference was added.
In general, it’s nice to end an introduction section with a few comments about what the review contains, to help readers understand what’s coming; please consider adding some relevant remarks here.
I followed your suggestion by inserting a sentence in the introduction that anticipates the topics included in the review.
It would be great to have a table with detailed information on the past and present therapies used, the mechanism targeted, outcomes, and limitations against SS.
Thanks for your suggestion, but I would like to clarify that it was not possible in this context to reuse in a more specific way the past and future therapies used for SS patients because it is a topic too extensive to be summarized in this context. This could be a topic that we may include in a future review. However, a generic table related to SS therapeutic strategy was added to the manuscript.
The schematics presentations by the authors made it attractive and easy for the readers to grasp.
We thank the reviewer very much for her/his positive comment and are pleased that the figures included in the review can make the manuscript easier to read.
There are a lot of definite and indefinite articles missing and is required to correct these. Please correct the manuscript for language and grammatical mistakes.
We revised the manuscript trying to improve the correction of the English language.
Round 2
Reviewer 1 Report
Thank for you making the corrections as suggested. I have no further comments.
Author Response
Thank you so much for your constructive and positive comments.